# The emotion regulation effect of cognitive control is related to depressive state through the mediation of rumination: An ERP study

**Shuzhen Gan**[1], **Shuang Chen**[2], **Xiangrong Shen**[3,4]*

**1** Lab for Psychological Health and Imaging, Shanghai Mental Health Center, Shanghai Jiao Tong University School of Medicine, Shanghai, China, **2** Department of Psychology, Zhejiang Normal University, Jinhua, China, **3** Key Laboratory of Behavioral Science, Institute of Psychology, Chinese Academy of Sciences, Beijing, China, **4** College of Humanities, Shanghai Normal University, Shanghai, China

* sxr@shnu.edu.cn

**Data Availability Statement:** All relevant data are within the manuscript and its Supporting Information files.

**Funding:** The authors received no specific funding for this work.

## Abstract

Deficits in cognitive control have been found in depression, but how they contribute to depressive symptoms remains unknown. The present study investigated whether the regulatory efficacy of cognitive control on negative emotion varies with depression level and whether the regulatory efficacy affects depressive symptoms via the mediation of rumination. Fifty participants screened by the Zung Self-Rating Depressive Scale (SDS) with high and low depression levels were selected. They were instructed to controlled-process different semantic representations of aversive pictures, and the amplitude of the late positive potential (LPP) evoked by the pictures was used as the measure of electrocortical response. We found that controlled-processing neutral representations of aversive pictures significantly decreased the amplitude of LPP relative to that under controlled-processing unpleasant ones in an early window in the low depression group and that this regulatory effect was impaired in the high depression group. Furthermore, a mediation analyses indicated that the regulatory efficacy of controlled-processing different semantic representations was associated with SDS score via the mediation of rumination. These findings shed light on the mechanisms underlying the association between the function of cognitive control in emotion generation and depressive symptoms and indicated a pathway from the regulatory efficacy of cognitive control to depression via rumination.

## Introduction

Depression is a highly prevalent mood disorder mainly characterized by sustained negative affect and anhedonia. Abnormalities in down-regulating negative emotions is a hallmark of depression; both clinically depressed patients and nonclinical individuals with high depression levels have difficulties in employing adaptive strategies such as cognitive reappraisal and have an increased tendency to use maladaptive strategy rumination relative to non-depressed controls or individuals with low depression levels [1, 2]. Cognitive models have proposed several risk factors of depression. For example, depressed patients and individuals who score high on

**Competing interests:** The authors have declared that no competing interests exist.

depression typically have negative attention, memory and interpretation biases toward events in daily life [3, 4]. However, how cognitive factors contribute to depressive symptoms is fully understood. Here we focused on cognitive control which is associated with the efficacy of emotion regulation [5]. Cognitive control has been found to be impaired in individuals with depressive symptoms, with such individuals having difficulty focusing attention on goal-relevant positive or neutral information and inhibiting interference from irrelevant negative information in social situations [6]. These deficits may impede individuals from reducing their negative emotions and lead to the accumulation of negative mood, the enhancement of rumination tendency, and the development or maintenance of depressive symptoms [6]. The current study sought to investigate how deficit of cognitive control contributes to depression by examining how the regulatory effect of cognitive control on negative emotion varies with depression level and the role of rumination in influencing the relationship between the regulatory efficacy of cognitive control and depressive symptoms.

Cognitive control involves a class of controlled processes in working memory (WM) to ensure the accomplishment of current task goals, including focusing attention on task-relevant information and inhibiting and updating irrelevant information [7]. Researchers have proposed that processes of cognitive control support emotion regulation [8] and alter emotional responses [9]. For instance, a commonly used and recommended strategy, cognitive reappraisal, relies on the processes of cognitive control to reduce unwanted emotions. Reappraisal involves understanding the emotional event from an alternative perspective (e.g., with less negative meaning) [10], in which controlled attention on the positive or neutral meaning and inhibition of the negative meaning of the event are indispensable. Neuroimaging studies have provided evidence that the regions of the cognitive control dorsolateral prefrontal cortex (dlPFC), ventrolateral prefrontal cortex (vlPFC) and dorsal anterior cingulate cortex (dACC) [11] are usually recruited by reappraisal [12]. Furthermore, correlation studies have revealed that cognitive control abilities are positively related to the effectiveness of reappraisal [5]. Consequently, individuals with a worse ability of cognitive control usually have more negative and less positive emotions in their daily life than those with better cognitive-control ability [13]. By directly demonstrating that controlled attention to different visual representations of negative pictures in WM changed the emotion-related electrocortical response, Thiruchselvam and colleagues (2012) revealed how processes of cognitive control influence the emotional response [14]. However, until now, no study has directly examined how controlled-processing different semantic representations of an emotional event influences the emotion outcome. Such studies are important for understanding the mechanisms underlying strategies based on semantic changes, such as reappraisal.

Though studies have suggested a possible role of cognitive control in regulating maladaptive emotions, unfortunately, various abilities of cognitive control have been demonstrated to be impaired to some degree in both clinically depressed patients and nonclinical individuals with high levels of depressive symptomatology. For instance, depressed patients needed more time than healthy controls to discard negative faces and words from WM [15, 16]. Individuals who score high on self-reported measures of depression exhibit impaired attention disengagement from negative information [17], a decreased ability to inhibit the impact of emotional faces [18] and negative words [19]. These findings suggest that the cognitive control deficit in such individuals not only occurs concurrently with depressive symptoms but also is a vulnerability factor of depression.

Neuroimaging studies have found that the impaired ability of cognitive control in both depressed and depression-prone people is related with the decreased function in the frontal regions such as inferior frontal gyrus (IFG) which are overlapped with the regions responsible for emotion regulation in both of depressed and depression-prone people [20, 21]. And the

abnormal connectivity between dlPFC and amygdala in participants with higher depressive symptoms implies the dysfunctional regulatory effect of cognitive control on emotion [22]. These results provide preliminary and indirect evidence for a possible role of cognitive deficits in emotion regulation dysfunction in depression. If cognitive control helps down-regulate negative emotions, people with depressive symptoms might have decreased efficacy of emotion regulation when using processes of cognitive control, thereby having accumulated and sustained negative emotions, facilitating the development of depression. However, this possibility remains to be verified.

In addition to resulting in decreased efficacy of emotion regulation, cognitive control deficits may lead to rumination, a featured strategy usually used by depressed people as well as nonclinical individuals with depressive symptoms in daily life [23–25]. Rumination involves uncontrollable repetitive and passive self-reflection focusing on negative thoughts and depressed mood, and it is an important factor underlying the onset, maintenance and recurrence of depression [26, 27]. It has been found that a decreased inhibiting ability contributes to future increased use of rumination and increased depressive symptoms [28]. Cognitive control deficits might prompt rumination by facilitating the activation and maintenance of negative representations in WM, thereby maintaining a depressive state [29]. Recent studies have found that rumination might be a mediator between cognitive factors and depressive symptoms [30]. Consistent evidence has emerged from cognitive training studies reporting that the effect of cognitive training on the alleviation of depressive symptoms is mediated by the attenuation of rumination after training [31, 32]. In these studies, self-reported or task-measured ability of cognitive control was measured as an independent variable or predictor; however, as emotion regulation difficulty is a hallmark of depression, the regulatory function of cognitive control on emotion can serve as a more direct and explanatory indicator for investigating the mechanism underlying the link between cognitive control and depressive symptoms.

The present study investigated how the regulatory efficacy of controlled-processing different semantic representations in WM on emotional response varies with depression level and whether rumination serves as a mediator in the association between the regulatory efficacy of cognitive control and depression. Participants with high and low depression levels were presented with an aversive picture and two different words that described neutral and unpleasant contents of the picture. The participants were instructed to memorize one of the two words for later recognition and discard the other one. We predicted that when controlled-processing different semantic representations (unpleasant vs. neutral) of the aversive picture in WM, the emotion outcome would be changed.

Event-related potentials (ERPs) were measured. Electroencephalogram (EEG) data were recorded and analyzed when the aversive pictures were presented. Studies in the emotion and emotion-regulation fields have focused on late positive potential (LPP) component of ERP, a positive deflection usually enhanced by emotional stimuli relative to neutral ones that covaries with the arousal of the stimuli [33]. The LPP is evident as early as 300–400 ms after stimulus onset and can last for seconds during the entire time window of the stimulus presentation [34]. It is considered to reflect attention to the emotional stimuli for their intrinsic motivational significance [35] and further elaborated processes [36, 37]. Various studies have found that the emotion-related LPP is sensitive to basic cognitive processes and complex emotion regulation strategies. For example, selectively focusing attention on the unemotional aspects of aversive pictures attenuates the amplitude of the LPP [38]. Reappraising emotional pictures from a negative or a neutral description also regulates LPP amplitude [39]. These findings suggest that emotion-related LPP is a reliable electrophysiological marker of the regulatory result of cognitive control.

In the current study, we predicted that controlled-processing unpleasant and neutral semantic representations in WM would produce different amplitudes of picture-evoked LPP

in the participants with low depression level and that this effect would be impaired in participants with high depression level. In addition, we predicted that the regulatory efficacy of cognitive control would correlate with depression level through the mediation of rumination.

## Materials and methods

### Participants

We recruited 120 college students from Shanghai Normal University through online advertisement. All of them completed a questionnaire, Zung Self-Rating Depression Scale (SDS, Chinese version) [40] to assess the severity of depressive symptoms. To select participants with high and low depression levels for subsequent ERP experiment, we used a cutoff score of 50 as people with scores greater than or equal to 50 have been rated as having depressive symptoms [41]. Finally, 50 students volunteered to participate the ERP experiment, comprising 26 participants with scores above and equal to 50, forming the high depression group, and 24 with scores below 50, composing the low depression group.

Demographic variables, such as gender and age were counterbalanced between these two groups. The characteristics of the sample are shown in Table 1. All participants were right-handed and reported normal or corrected-to-normal vision. The exclusion criteria for both groups included a history of drug addiction, alcohol addiction, medication use within the last two weeks, and neurological or psychiatric disorders. Participants were paid 20 yuan after filling out the SDS questionnaire and 80 yuan after completing the ERP experiment and psychometric testing for rumination.

This study was approved by the ethics committee of Shanghai Normal University. All participants provided written informed consent in accordance with the Declaration of Helsinki.

### Psychometric tests

**Measure of depression.** The SDS includes 20 items measuring affective, psychological, and physical symptoms related to depression. Volunteers rated each item concerning how often he or she experienced each symptom within a week on a scale from 1 (none or a little of the time) to 4 (most or all of the time). The SDS has a score range of 20–80. According to Zung's studies, scores less than 50 are in the normal range, and scores of 50–59, 60–69 and above 70 indicate mild, moderate and severe depressive symptoms, respectively [41]. In the high depression group, participants' scores on SDS were all greater than or equal to 50 (min = 50, max = 65). In the low depression group, the scores were all below 50 (min = 28, max = 46). The SDS scores of the two groups were significantly different ($t(48) = 12.27$, $p < 0.001$, *Cohen's d* = 3.55).

**Measure of rumination tendency.** The Ruminative Response Scale (RRS) [42] contains 22 items assessing the degree people to which a person uses rumination to cope with

**Table 1. Demographic variables of the participants in the high and low depression groups.**

|  | High depression | Low depression |
|---|---|---|
| N (female) | 26 (22) | 24 (21) |
| Age (*M* ± *SD*) | 23.12 ± 1.34 | 23.29 ± 1.23 |
| RRS (*M* ± *SD*) | 47.77 ± 7.67 | 34.13 ± 5.50 |
| SDS (*M* ± *SD*) | 53.60 ± 3.88 | 36.56 ± 5.61 |

N = number, M = mean, SD = standard deviation, RRS = ruminative response scale, and SDS = self-rating depressive scale.

depressive contents in his or her daily life. The participants responded to the items on a scale ranging from 1(almost never) to 4 (almost always), where a higher score suggests a higher engagement in rumination. The rumination scores in the high depression group (min = 30, max = 63) were higher than the scores in the low depression group (min = 24, max = 58) ($t$(48) = 7.17, $p <$ 0.001, *Cohen's d* = 1.04).

**Stimuli.** In total, 135 pictures (90 aversive and 45 neutral) from the International Affective Picture System (IAPS) [43] were used in the experiment. According to the normative ratings of Lang et al. (2005), aversive pictures ($M$ = 2.98, $SD$ = 0.91) were rated as more unpleasant than neutral pictures ($M$ = 6.33, $SD$ = 1.04) [40]. In addition, the emotional arousal of aversive pictures ($M$ = 5.42, $SD$ = 0.88) was higher than that of neutral pictures ($M$ = 3.83, $SD$ = 1.22).

To examine whether the pictures used in the present study had similar emotional significance to Chinese participants as to a Western population, another 50 college students were recruited. They were required to rate each picture on two dimensions, valence and arousal, on a rating scale from 1 (completely unhappy or melancholic or valence; completely relaxed or calm for arousal) to 9 (completely happy or pleased for valence; completely excited or aroused for arousal). We found that aversive pictures were rated as more unpleasant ($M$ = 2.95, $SD$ = 0.75) than neutral pictures ($M$ = 5.67, $SD$ = 0.54) ($t$ (49) = 19.43, $p <$ 0.001, *Cohen's d* = 2.75). In addition, aversive pictures were rated as more arousing ($M$ = 6.21, $SD$ = 0.99) than neutral pictures ($M$ = 3.42, $SD$ = 0.76) ($t$ (49) = 18.25, $p <$ 0.001, *Cohen's d* = 2.58). These results suggest that the selected aversive and neutral pictures differ in the valence and arousal dimensions in Chinese participants.

In addition, a total of 135 pairs of words (270 words in total) were used, and each pair was associated with a picture. For the aversive pictures, the two words in the pair provided contents with different valence about the situation depicted in the picture, with one word highlighting the unpleasant content and the other focusing on neutral or emotion-attenuated content (e.g., "curse" and "wrinkle" for an image in which an old man is angrily saying something). In all, 90 pairs of words were used when aversive pictures were presented. Another 45 pairs of words were employed for the neutral pictures; both words in each pair highlighted neutral content (e.g., "horse" and "grassland" for an image in which a horse is running on the grassland). All the words were of high frequency in the Chinese language, and each consisted of two Chinese characters.

**Task design and procedure.** The participants were required to complete a recognition task. In the trials of aversive pictures, the two words describing the unpleasant and neutral contents of the previously displayed aversive picture were concurrently presented to the participants. A red dot then appeared on either of the locations of the two words to indicate the word that was relevant for later recognition. The participants were asked to focus attention on and remember the relevant word and discard the irrelevant word to the greatest extent possible after the cue dot appeared. After the picture was displayed later in a trial, the participants decided whether the probe word was the relevant word. In this recognition stage, the probe would be the relevant word, the irrelevant, or a completely new word. The frequency of presentation was equal among these three types of probe. Thus, six types of words in the recognition stage were used: relevant-unpleasant, relevant-neutral, irrelevant-unpleasant, irrelevant-neutral, new-unpleasant and new-neutral. In the trials with neutral pictures, both words were neutral, and participants were instructed to process either of them and complete the recognition task. Three types of words were involved: relevant-neutral, irrelevant-neutral and new-neutral.

The task was designed to influence the picture processing. Following the instruction stage, the participants were expected to manipulate the semantic representations of the picture in WM, and two controlled-processing (CP) conditions were involved: controlled-processing the semantic representation of the neutral content (CP-N) and controlled-processing the representation of the unpleasant content (CP-U). Under the CP-N condition, the dot appeared in

the place of neutral word, and the participants would elaborate and maintain the goal-relevant neutral word in WM and resolve the interference from the irrelevant unpleasant word, for example, by inhibiting the processing on it and decreasing its activation in WM. Consequently, the neutral semantic representation would be integrated into the further processing of the aversive picture, and the emotion-enhanced LPP to the aversive picture would be influenced. In contrast, under the CP-U condition, the dot appeared in the place of the unpleasant word; the unpleasant semantic representation would be integrated into the processing of the picture and the picture-evoked LPP would be increased relative with that in the CP-N condition. In the trials of neutral pictures, the two presented words were both neutral, and participants would process either of them according to the dot location (NEU). Therefore, three conditions were involved in picture processing: CP-U, CP-N and NEU.

A typical trial is shown in Fig 1. The trial started with a fixation, which was displayed for a duration jittered to be 1000, 1250, or 1500 ms (average interval = 1250 ms). After a picture was displayed for 2000 ms, the word pair appeared for 3000 ms. Subsequently, a red dot was displayed for 1000 ms, followed by another duration jittered to be 1000, 1250, or 1500 ms (average interval = 1250 ms). After the picture was re-displayed for 2000 ms, the probe word was presented in the center of the screen. The participants were instructed to decide whether the probe word was the target word within 3000 ms. They were instructed to press the key "j" for a

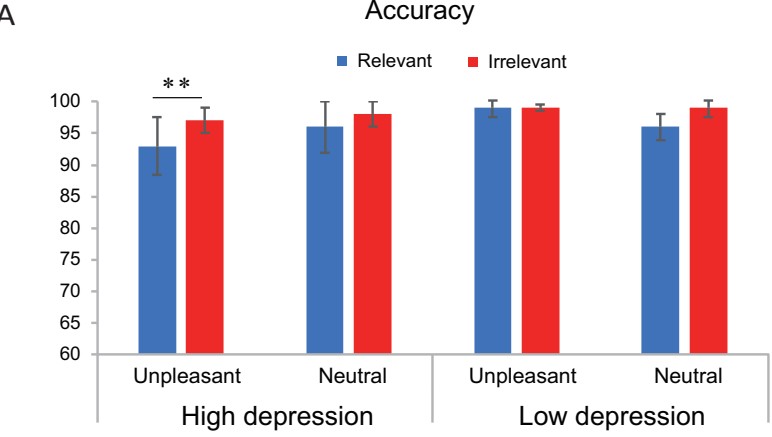

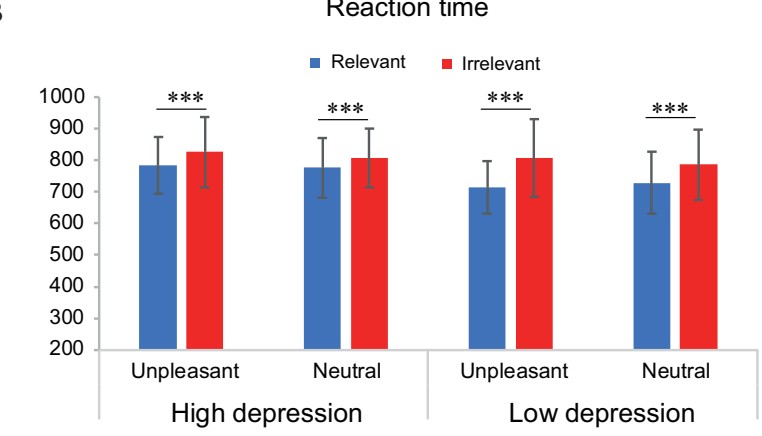

**Fig 1. The display sequence and timing in the trials.**

yes response and the key "f" for a no response as quickly and as accurately as possible. The next trial was then started. The inter-trial interval (ITI) was jittered to be 1000, 1250, 1500, 1750, or 2000 ms (average ITI = 1500 ms). The words with different valences and the cue dot appeared randomly at the left and right locations an equal number of times. In total, 90 trials for aversive pictures and 45 trials for neutral pictures were included in the experiment. The trials of aversive pictures and neutral pictures were intermixed, and the sequence of trials was randomized.

Pictures and words were presented on a 23-inch screen with a 60-Hz refresh rate and 200 cd/m$^2$ brightness. E-Prime software (Psychological Software Tools, Inc., Pittsburgh, Pennsylvania, USA) was used for the presentation of stimuli and response recording. The visual angle of each picture and word pair was approximately 40° horizontally and vertically, with each picture or word pair occupying approximately 80% of the screen.

After the participants provided informed consent, they were led into a separate room and seated in front of a computer monitor at a distance of 60–80 cm. They were then fitted with an EEG recording cap and guided through 20 practice trials, which included pictures and words different from those used in the experiment. After becoming familiar with the task, the participants proceeded to the experimental procedure, which was divided into three sessions, each containing 45 trials. The participants were allowed a rest period in the interval between sessions to ensure they could focus on the experimental task attentively. Moreover, a rest screen was displayed in the middle of each session, allowing the participants to control the break time by themselves. The response accuracy and reaction time to the probe word were recorded in each trial. The task lasted for approximately 40 min. After completing the ERP experiment, all participants filled out the RRS immediately.

**EEG data recording and preprocessing.** Continuous EEG data were recorded using SynAmps amplifiers and were digitized using Curry 7 (Neuroscan, Charlotte, North Carolina, USA). Sixty-four Ag/AgCI electrodes were used in the cap, and they were placed according to the 10/20 system. An electrode placed on the right mastoid served as an online reference. Vertical electrooculogram (EOG) signals generated from eye blinks were recorded from sites 1 cm above and below the left eye, and horizontal EOG signals were recorded from the bilateral external canthus. The EEG signal was sampled at a rate of 1000 Hz. All electrode impedances were kept below 10 kΩ.

The EEG data of aversive and neutral pictures presented the first and second time in each trial were preprocessed with the same procedure using Neuroscan 4.5 software. Vertical ocular artifacts were removed from the EEG signals with a regression procedure devised by Semlitsch et al. [44]. Next, the EEG data were filtered at 0.01–30 Hz (24 dB/octave) and were segmented to epochs starting from 500 ms before the picture onset, continuing for the entire duration of picture presentation (2000 ms). Next, the 500 ms pre-stimulus recording was used for baseline correction. Trials with excessive physiological artifacts exceeding ±100 µV were discarded; however, over 90% of the original trials remained for further analyses. All EEG data were re-referenced to the average of the left and right mastoids.

## Data analyses

**Behavioral data.** We checked the performance of controlled-processing different words (unpleasant vs. neutral) associated with the aversive pictures. The behavioral measures in the word recognition stage between the relevant and irrelevant words were compared in different emotions and groups. Three-way analysis of variance (ANOVA) of accuracy and reaction time was performed with the factors depression level (high and low depression), probe type (relevant and irrelevant) and probe valence (unpleasant and neutral).

**ERP data.** To examine the emotion effect in picture processing on the amplitude of LPP, we compared the amplitude of the LPP elicited by the aversive pictures with that elicited by the neutral ones for pictures presented for the first time in the trial. For each participant, the ERPs were averaged according to picture type (aversive and neutral). Previous studies suggested that the LPP amplitude elicited by emotional stimuli is maximal at central-parietal sites and may differ from that at anterior sites [45]; thus, the electrodes were divided into two regions of interest: an anterior region, containing data from ten anterior electrodes (F1, F2, F3, F4, Fz, FC1, FC2, FC3, FC4, FCz), and a parietal region, containing data from fifteen parietal electrodes (C1, C2, C3, C4, Cz, CP1, CP2, CP3, CP4, CPz, P1, P2, P3, P4, Pz). In each of these two regions, ERP data were averaged across electrodes. The LPP amplitudes were averaged in each of three time windows: 500–1000 ms (early), 1000–1500 ms (middle), and 1500–2000 ms (late). A three-way repeated-measures ANOVA with the factors depression level (high and low depression), picture type (aversive and neutral), and site (anterior, parietal) was conducted for each time window.

Similar statistical methods were used to investigate whether controlled-processing different semantic representations (unpleasant vs. neutral) of the aversive pictures modulated the emotion-enhanced LPP in the two groups. A three-way repeated-measures ANOVA with the factors depression level (high and low depression), CP (CP-U, CP-N, NEU) and site (anterior, parietal) was conducted for the early, middle and late window.

All analyses were performed using SPSS version 20.0 (IBM, Armonk, NY). Greenhouse-Geisser corrections were applied when the sphericity hypothesis was violated. Bonferroni corrections were used for multiple-comparison tests. Data from one participant were identified as outlier and was removed, as the ERP amplitudes in all conditions were larger than 3 standard deviations from the mean.

**Linear regression and mediation analyses.** To test whether the regulatory effect of cognitive control on picture-evoked LPP is correlated with depression level and rumination tendency, linear regressions were conducted with data from the early, middle and late windows. To investigated whether rumination is a mediator of the relationship between the regulatory function of cognitive control and depressive symptoms, mediation analyses using the PROCESS macro for SPSS [46] were performed for the three windows. In all three models, the amplitude change of LPP by cognitive control was the independent variable, depression level was the dependent variable, and rumination was the hypothesized mediator (Fig 2). The direct path from the emotion regulatory effect of cognitive control to depression level and the indirect path from the cognitive control to depression via the rumination were investigated. Analyses for indirect effects were conducted with 10000 bootstrap samples, and bias-corrected bootstrap confidence intervals were calculated.

## Results

### Behavioral data

The mean accuracy and reaction time for the different conditions are shown in Fig 3. Statistical analysis indicated that participants responded to irrelevant words more slowly than to relevant words ($F(1,47) = 15.31$, $p < 0.001$, $\eta_p^2 = 0.25$), which demonstrated the effect of cognitive control. Differences in responding latency between the unpleasant and neutral words and the differences between the high and low depression groups were not observed ($ps > 0.05$). Neither a two-way nor a three-way interaction was found in the statistical analyses of reaction time ($ps > 0.05$).

The analysis of accuracy revealed significant effects of probe type ($F(1,47) = 5.30$, $p < 0.05$, $\eta_p^2 = 0.10$) and depression level ($F(1,47) = 4.00$, $p = 0.05$, $\eta_p^2 = 0.08$). A two-way interaction between depression level and probe valence ($F(1,47) = 13.36$, $p < 0.001$, $\eta_p^2 = 0.22$) and a

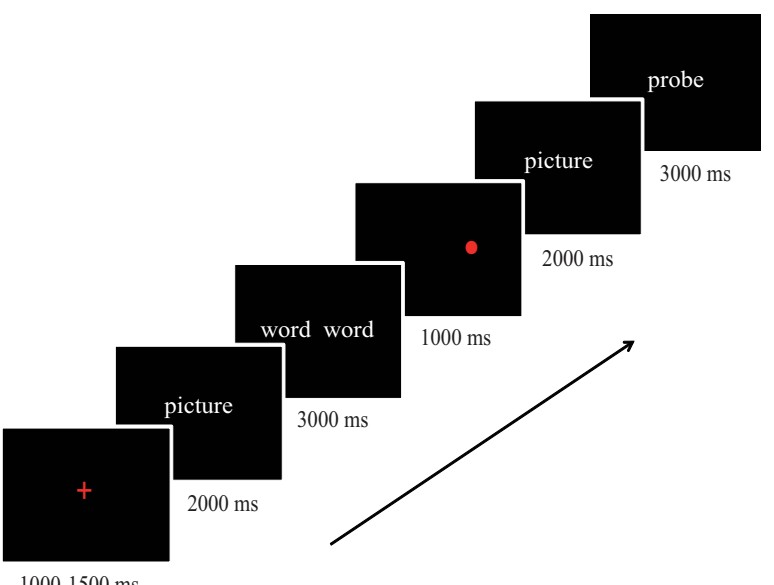

**Fig 2. Model of mediation analyses.**

three-way interaction among the depression level, probe type and probe valence ($F(1,47) =$ 4.13, $p < 0.05$, $\eta_p^2 = 0.08$) were found. A simple effect analysis indicated that participants in the high depression group responded to the unpleasant irrelevant words more correctly than to the unpleasant relevant words ($F(1,47) = 8.71$, $p < 0.01$, $\eta_p^2 = 0.14$). A difference between the irrelevant and relevant word for the unpleasant type in the low depression group or for the neutral type in either group was not found ($ps > 0.05$).

### ERP data

The grand-average waveforms for the aversive and neutral pictures presented the first time in the trial in the high and low depression groups are displayed in Fig 4. The grand-average waveforms for the three controlled-processing conditions when the second pictures were presented

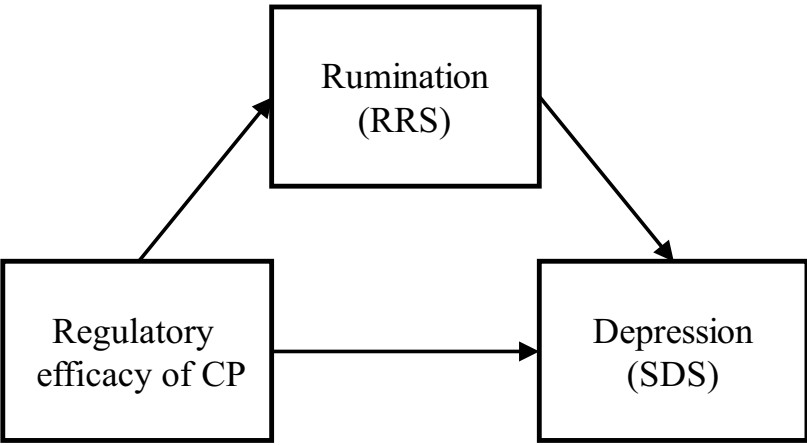

**Fig 3.** Mean accuracy (A) and reaction time (B) of recognition for the probe words in different conditions. Error bars indicate standard deviations. ** $p < 0.01$, *** $p < 0.001$.

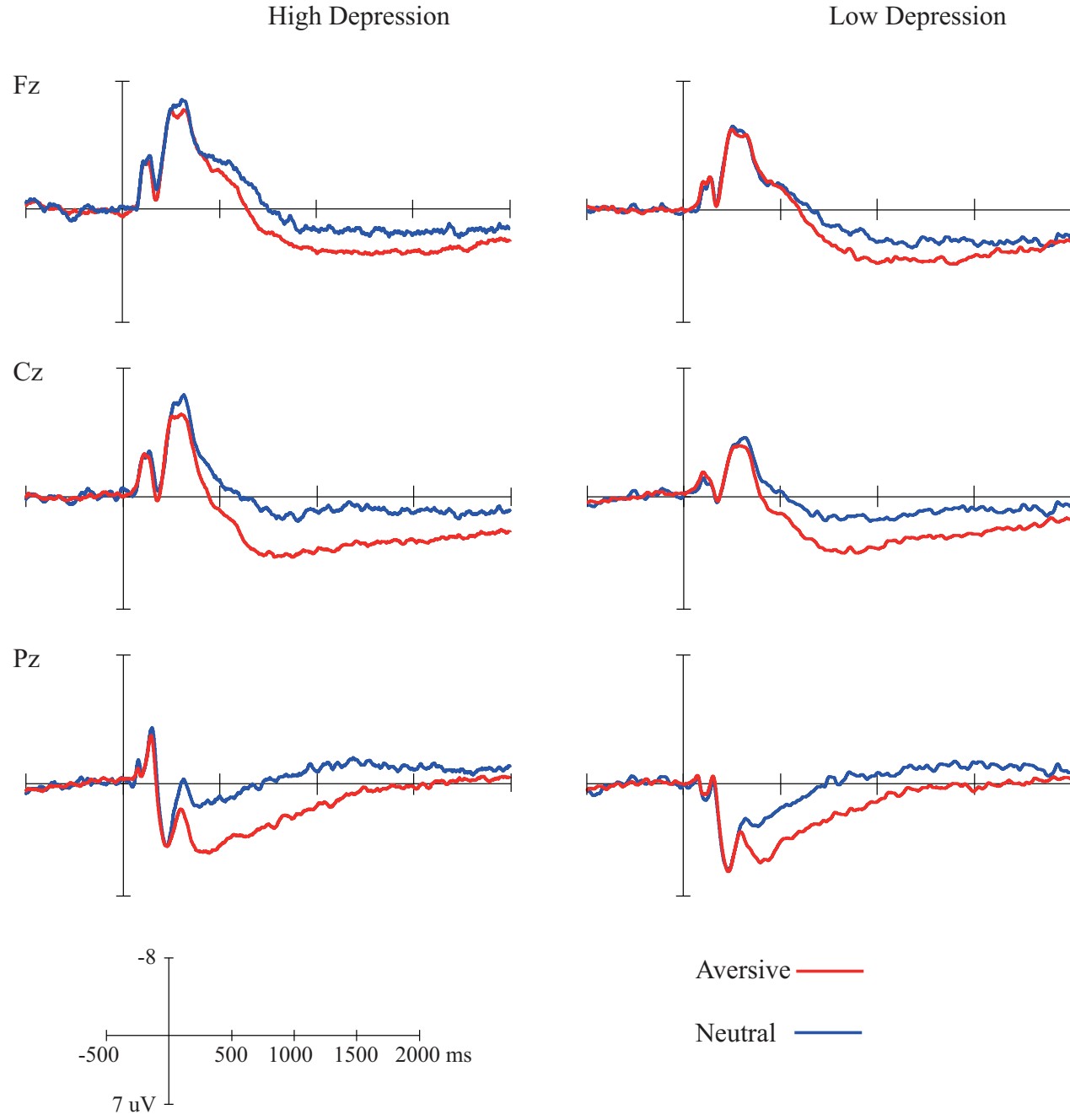

**Fig 4.** Grand average of ERPs for the pictures presented for the first time in the high (left) and low (right) depression groups. Waveforms at Fz, Cz, and Pz are depicted.

in the two groups are shown in Fig 5. The mean LPP amplitudes and standard deviations are displayed in Table 2.

### Emotion effect

An effect of picture emotion on LPP amplitude was found in all three time windows (early: $F(1,47) = 66.06$, $p < 0.001$, $\eta_p^2 = 0.58$; middle: $F(1,47) = 53.49$, $p < 0.001$, $\eta_p^2 = 0.53$; late:

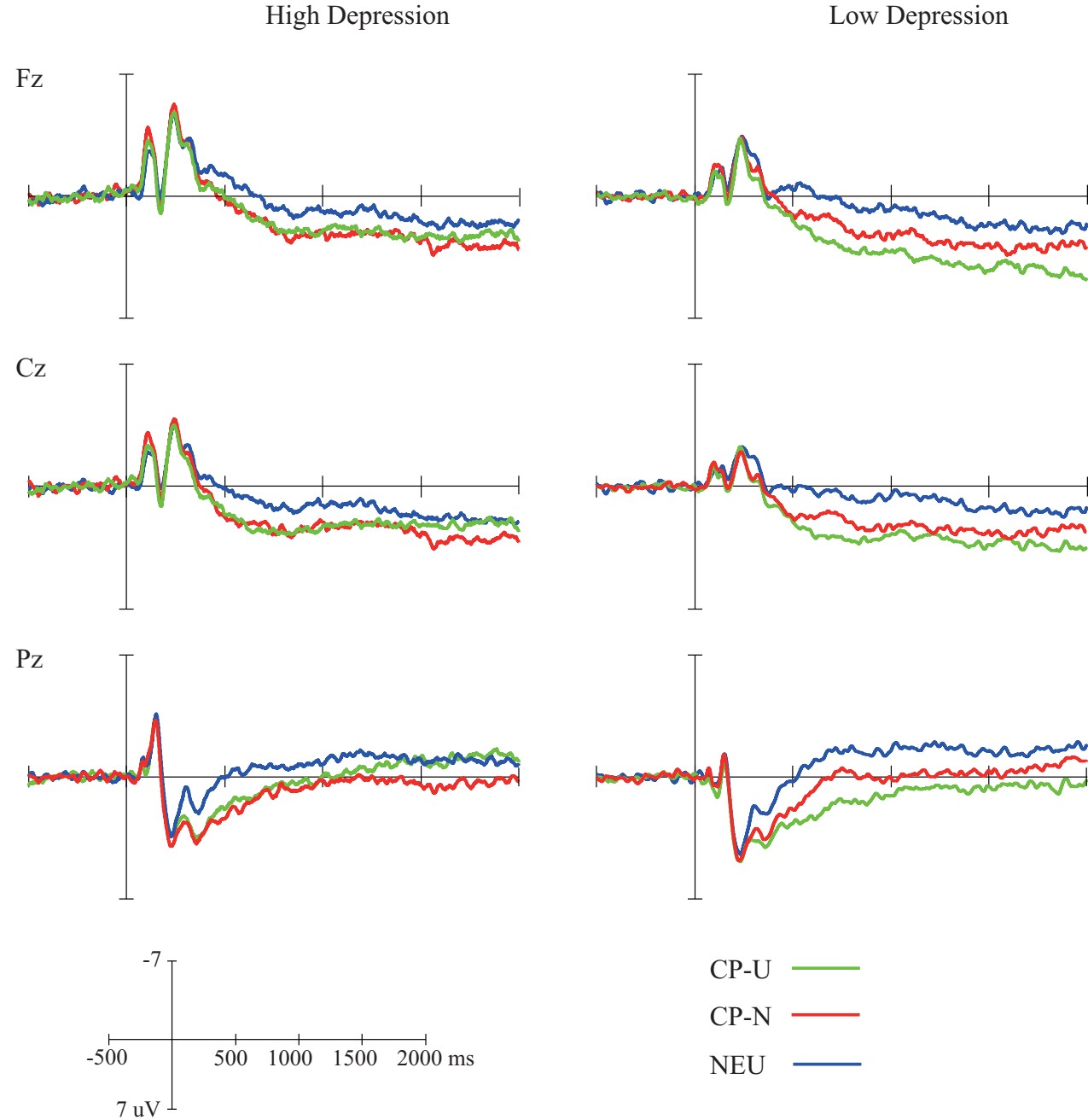

High Depression                    Low Depression

CP-U
CP-N
NEU

**Fig 5.** Grand average of ERPs for the pictures presented the second time in the high (left) and low (right) depression groups. Waveforms at Fz, Cz and Pz are depicted.

$F(1,47) = 17.3$, $p < 0.001$, $\eta_p^2 = 0.27$), with aversive pictures eliciting significantly larger LPPs than neutral ones. In addition, the picture-evoked LPPs had larger amplitudes in parietal electrodes than in the anterior electrodes (early: $F(1,47) = 15.71$, $p < 0.001$, $\eta_p^2 = 0.25$; middle: $F(1,47) = 15.61$, $p < 0.001$, $\eta_p^2 = 0.0.25$; late: $F(1,47) = 19.32$, $p < 0.001$, $\eta_p^2 = 0.29$), which is consistent with previous studies [42]. An interaction between picture emotion and site was observed in the early window ($F(1,47) = 23.90$, $p < 0.001$, $\eta_p^2 = 0.34$), and the simple effect analyses indicated that aversive pictures enhanced LPP amplitude in all anterior and parietal

**Table 2. Mean LPP amplitudes and standard deviations (μV) for pictures presented the first and second time under different conditions.**

| | | High depression | | | Low depression | | |
|---|---|---|---|---|---|---|---|
| | | **Early** | **Middle** | **Late** | **Early** | **Middle** | **Late** |
| **First picture** | | | | | | | |
| Aversive | anterior | 1.57 (2.91) | 3.16 (2.90) | 2.93 (3.19) | 1.70 (3.22) | 2.98 (3.26) | 2.50 (3.83) |
| | parietal | 3.71 (2.57) | 2.35 (2.28) | 1.64 (2.85) | 3.16 (3.96) | 1.92 (3.20) | 1.30 (3.43) |
| Neutral | anterior | 0.13 (2.66) | 1.79 (3.01) | 1.85 (2.93) | 0.58 (2.62) | 1.78 (2.93) | 1.72 (3.48) |
| | parietal | 1.26 (2.30) | 0.63 (2.23) | 0.73 (2.67) | 1.26 (2.53) | 0.53 (2.61) | 0.37 (3.07) |
| **Second picture** | | | | | | | |
| CP-N | anterior | 1.02 (2.93) | 1.76 (2.89) | 2.10 (3.16) | 0.90 (2.80) | 1.70 (3.18) | 1.63 (3.26) |
| | parietal | 2.32 (2.17) | 1.63 (2.69) | 1.75 (3.11) | 1.58 (2.09) | 1.23 (2.31) | 0.91 (2.75) |
| CP-U | anterior | 0.95 (3.37) | 1.83 (3.07) | 1.79 (3.05) | 1.56 (2.99) | 2.53 (3.13) | 2.59 (3.59) |
| | parietal | 2.26 (2.18) | 1.33 (2.74) | 0.89 (3.43) | 2.55 (2.37) | 1.89 (2.69) | 1.68 (3.14) |
| NEU | anterior | -0.28 (2.89) | 0.73 (2.91) | 1.08 (3.23) | -0.08 (3.66) | 0.81 (3.45) | 0.92 (3.63) |
| | parietal | 0.86 (2.33) | 0.47 (2.86) | 0.72 (3.34) | 0.36 (3.17) | 0.08 (3.08) | 0.14 (3.29) |

CP-N = controlled-processing neutral word for the aversive picture, CP-U = controlled-processing unpleasant word for the aversive picture, and NEU = controlled-processing neutral word for the neutral picture.

electrodes (anterior: $F(1,47) = 33.26$, $p < 0.001$; parietal: $F(1,47) = 82.48$, $p < 0.001$). A group effect was not observed in any of the three windows ($ps > 0.05$), and the effects of two- and three-way interactions of group with emotion and site did not reach significance ($ps > 0.05$).

### Controlled-processing (CP) effect

A significant CP effect on aversive picture-evoked LPP was found in all three time windows (early: $F(2,94) = 26.38$, $p < 0.001$, $\eta_p^2 = 0.36$; middle: $F(2,94) = 14.11$, $p < 0.001$, $\eta_p^2 = 0.23$; late: $F(2,94) = 6.21$, $p < 0.01$, $\eta_p^2 = 0.12$). Furthermore, parietal electrodes had more positive LPP than anterior electrodes in the early ($F(1,47) = 7.56$, $p < 0.01$, $\eta_p^2 = 0.14$) and late window ($F(1,47) = 4.55$, $p < 0.01$, $\eta_p^2 = 0.09$). An effect of group and two- and three-way interactions of group with CP and site were not found ($ps < 0.05$).

Planned multiple comparisons were performed among CP-U, CP-N and NEU conditions in the anterior and parietal sites in different groups, and the statistical results are presented in Table 3. In the low depression group, controlled-processing unpleasant (CP-U) and neutral (CP-N) words led to a prominent difference in LPP amplitude evoked by aversive pictures in the early window (parietal: $t(22) = 3.56$, $p < 0.05$, Cohen's $d = 0.74$). This result suggested a regulatory effect of controlled-processing on emotional response. However, evidence of a regulatory effect was not found in the high depression group ($ps > 0.05$).

In addition, in the low depression group, larger amplitudes of LPP were found in controlled-processing unpleasant words when aversive pictures were presented (CP-U) than controlled-processing neutral words when neutral pictures were presented (NEU) in the early (anterior: $t(22) = 4.01$, $p < 0.01$, Cohen's $d = 0.84$; parietal: $t(22) = 4.54$, $p < 0.001$, Cohen's $d = 0.95$), middle (anterior: $t(22) = 4.41$, $p < 0.001$, Cohen's $d = 0.92$; parietal: $t(22) = 3.62$, $p < 0.05$, Cohen's $d = 0.76$) and late windows (anterior: $t(22) = 3.31$, $p < 0.05$, Cohen's $d = 0.69$). Such a difference between controlled -processing neutral words of aversive pictures (CP-N) and controlled-processing neutral words of neutral pictures (NEU) was not detected in the early, middle and late windows in the anterior electrodes ($ps > 0.05$). In the high depression group, CP-U condition had more positive LPPs than the NEU condition in the early window (anterior electrodes: $t(25) = 4.06$, $p < 0.001$, Cohen's $d = 0.8$, parietal electrodes: $t(25) =$

**Table 3. Results of t-tests for the CP-N, CP-U and NEU conditions.**

| | Early | | | Middle | | | Late | | |
|---|---|---|---|---|---|---|---|---|---|
| | *df* | *t* | *d* | *df* | *t* | *d* | *df* | *t* | *d* |
| **High depression** | | | | | | | | | |
| anterior CP-N vs. CP-U | 25 | -0.28 | — | 25 | 0.22 | — | 25 | -0.89 | — |
| anterior CP-U vs. NEU | 25 | 4.06*** | 0.8 | 25 | 3.10* | 0.61 | 25 | 1.52 | — |
| anterior CP-N vs. NEU | 25 | 3.60** | 0.71 | 25 | 2.22 | — | 25 | 2.26 | — |
| parietal CP-U vs. CP-N | 25 | -0.2 | — | 25 | -0.75 | — | 25 | -2.37 | — |
| parietal CP-U vs. NEU | 25 | 4.30*** | 0.84 | 25 | 2.36 | — | 25 | 0.38 | — |
| parietal CP-N vs. NEU | 25 | 4.45*** | 0.87 | 25 | 2.56 | — | 25 | 2.06 | — |
| **Low depression** | | | | | | | | | |
| anterior CP-U vs. CP-N | 22 | 1.94 | — | 22 | 1.87 | — | 22 | 1.68 | — |
| anterior CP-U vs. NEU | 22 | 4.01** | 0.84 | 22 | 4.41*** | 0.92 | 22 | 3.31* | 0.69 |
| anterior CP-N vs. NEU | 22 | 2.65 | — | 22 | 2.25 | — | 22 | 1.48 | — |
| parietal CP-U vs. CP-N | 22 | 3.56** | 0.74 | 22 | 1.77 | — | 22 | 1.62 | — |
| parietal CP-U vs. NEU | 22 | 4.54*** | 0.95 | 22 | 3.62* | 0.76 | 22 | 2.79 | — |
| parietal CP-N vs. NEU | 22 | 2.95* | 0.61 | 22 | 3.07* | 0.64 | 22 | 1.67 | — |

*p < 0.05

**p < 0.01

*** p < 0.001. The significance values of *t*-tests were all Bonferroni-corrected.

4.30, $p < 0.001$, *Cohen's d* = 0.84) and middle window (anterior electrodes: $t(25) = 3.10$, $p < 0.05$, *Cohen's d* = 0.61). The CP-N condition did not decrease the aversive picture-evoked LPPs relative to NEU condition evoked by neutral pictures in the early window in the anterior electrodes ($p > 0.05$).

## Regression and mediation analyses

As a regulatory effect of controlled-processing different representations of the aversive pictures was only observed in the parietal electrodes, the subsequent regression analyses were conducted with LPP amplitude in the parietal electrodes. As shown in Fig 6, the difference in LPP amplitude of CP-U vs. CP-N in each of the early, middle and late windows was negatively correlated with SDS score in all participants (early: $\beta = -0.30$, $t = -2.14$, $p < 0.05$; middle: $\beta = -0.32$, $t = -2.32$, $p < 0.05$; late: $\beta = -0.43$, $t = -3.29$, $p < 0.01$). In addition, the difference in LPP amplitude of CP-U vs. CP-N was negatively correlated with rumination tendency in all participants (early: $\beta = -0.47$, $t = -3.60$, $p < 0.01$; middle: $\beta = -0.49$, $t = -3.85$ $p < 0.001$; late: $\beta = -0.58$, $t = -4.93$, $p < 0.001$). These results suggested that the participants with higher rumination tendency had a worse regulatory effect of controlled-processing on emotion.

Three mediation models for the early, middle and late windows were tested. In each model, the effect of LPP change by cognitive control on depression was tested, with rumination as potential mediator. In all three windows, the modulation of LPP had no direct effect on depression level (early: $B = 0.18$, $t(46) = 0.23$, $p > 0.05$, 95% CI [-1.35 1.71]; middle: $B = 0.11$, $t(46) = 0.18$, $p > 0.05$, 95% CI [-1.11 1.34]; late: $B = -0.21$, $t(46) = -0.35$, $p > 0.05$, 95% CI [-1.38 0.97]). However, the indirect effect of the modulation of LPP on depression via rumination was prominent (early: $B = -2.06$, 95% CI [-3.62–0.90]; middle: $B = -1.71$, 95% CI [-2.76–0.87]; late: $B = -1.72$, 95% CI [-2.69–1.04]). These results indicated a complete mediation effect of rumination on the path from the regulatory function of cognitive control to depression level.

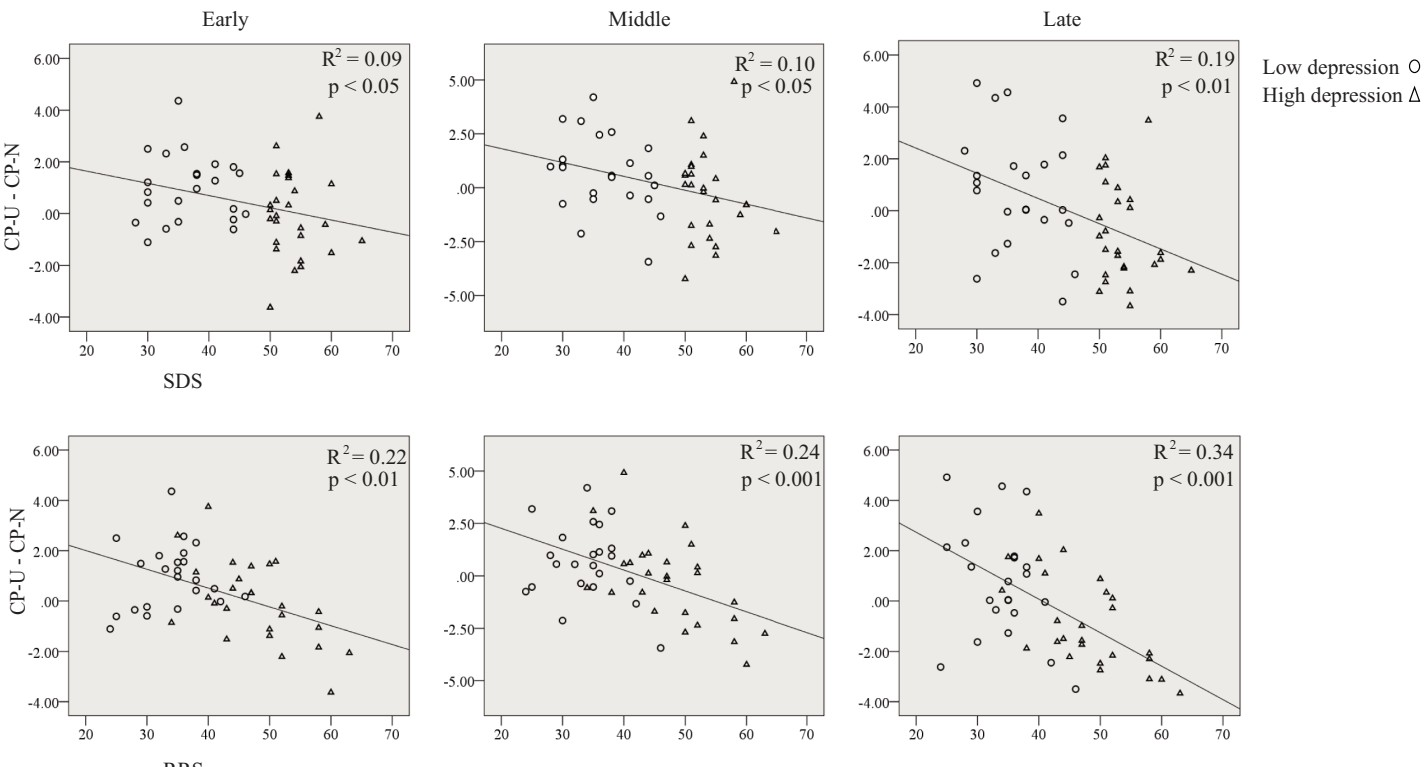

**Fig 6.** Scatter plots of depression (upper) or rumination (bottom) score and the difference in LPP amplitude between CP-U and CP-N conditions at parietal electrodes in the early, middle and late time window.

## Discussion

The present study investigated the emotion regulatory effect of cognitive control in people with different depression levels and how impaired regulatory efficacy contributes to depression. We found that controlled-processing the unpleasant semantic representation in WM significantly decreased aversive picture-evoked LPP relative to that under controlled-processing neutral ones in participants with low depression level, whereas this effect was impaired in participants with high depression level. Furthermore, we found that the regulatory function of cognitive control affected depression level via the mediation of rumination tendency.

Many studies have investigated the regulatory effects of complicated emotion regulation strategies such as reappraisal, which involves a set of cognitive processes. However, how the subprocesses involved in the strategies influence emotional response needed to be examined to promote our understanding of how effective regulatory strategies operate. Existing studies have found that executive attention to different visual representations in WM has a significant regulatory effect on emotion [14]. Our study further demonstrated that controlled-processing different semantic representations in WM modulate emotion-enhanced electrocortical responses. Specifically, after participants directed attention to the relevant neutral words and inhibited the processing of the irrelevant unpleasant words associated with the aversive pictures, their emotion-enhanced LPPs were decreased relative to those associated with controlled-processing unpleasant ones. The results shed light on the cognitive mechanism of reappraisal and are consistent with previous models positing that reappraisal is a linguistic and semantic strategy in which semantic representations of emotional events are changed to

decrease their emotional significance [47]. The results also provide cogent evidence supporting neuroimaging findings indicating that in reappraisal, cognitive control regions (e.g., PFC) indirectly influence amygdala responses to emotional stimuli via modulating semantic representations in the lateral temporal cortex [12]. In addition, a difference in LPP between controlled-processing of different words was found only in the early window; however, this does not necessarily mean that the regulatory effect of cognitive control on emotion disappeared in the middle and late windows. We found that controlled-processing neutral words of aversive pictures decreased the LPP amplitudes, and it led to a disappeared difference between aversive pictures and neutral pictures in each time window in anterior electrodes. These results suggest a regulatory effect of cognitive control on emotion.

The decrease in LPP amplitude after controlled-processing neutral words relative to that after controlled-processing unpleasant ones when aversive pictures were presented was absent in those participants with high depression level in the early window. The increased recognition accuracy for unpleasant irrelevant words than for unpleasant relevant ones confirmed the impaired ability to inhibit irrelevant negative representations in the participants with high depression level in our study. This finding is consistent with the finding of impaired ability of top-down attention control on mood-congruent distractors in individuals with social anxiety [48]. With such deficits, the negative representations will be activated and maintained in WM and occupy the limited resources of WM, and participants' emotional response will be difficult to modulate. Our results reveal the mechanisms underlying the relationship between cognitive control deficit and elevated negative emotions in depression [49] and provide evidence consistent with the impaired prefrontal-limbic emotion regulation circuit demonstrated in neuroimaging studies. LPP evoked by emotional pictures is considered to indicate downstream processes in the amygdala [45]. Consistently, amygdala activation is difficult to regulate in depressed individuals. In emotion regulation in healthy individuals, activation of the dorsolateral prefrontal cortex and the anterior cingulate cortex, which are involved in executive functions, is increased to downregulate the activities in the amygdala [50, 51, 52]. However, depressed individuals show a combination of dysfunction of the prefrontal regions, sustained high activation of the amygdala and abnormal connections among them [53], indicating difficulties in recruiting brain regions involved in cognitive control to regulate emotion-related neural responses.

Although controlled-processing the unpleasant meaning of aversive pictures elicited significantly larger LPPs than controlled-processing the neutral meaning of neutral pictures in the early window in both groups, we found that this enhancement lasted for a shorter time in the high depression group than in the low depression group. The decreased amplitude of LPP when controlled-processing unpleasant meaning might reflect active emotion suppression in the high depression group to avoid elevated negative emotion reactions when negative information was highlighted. This effect of emotion suppression in a late stage of emotion generation is consistent with previous studies with depressed participants [54]. It has been found that emotion suppression is one of the automatically and frequently used maladaptive strategies in depressed and depression-prone people and that emotion suppression is associated with decreased well-being [55]. Emotion suppression is a response-focused strategy involving attempts to reduce subjective feelings and physiological arousal after they have been initiated and has been demonstrated to have a downregulation effect on negative emotion in depressed people [56]. Therefore, the increased amplitude of LPP under the CP-U relative to that under NEU condition in the early window indicates an intact ability to detect and recognize stimuli with emotional salience. The gradual disappearance of this effect might reflect the effect of emotion suppression. In addition, the decreased amplitude under the CP-U condition led to a tendency of lower LPP in CP-U condition than in the CP-N condition in the middle and late

windows. In contrast to the difference between CP-U and CP-N in the early window, which reflected an emotion regulation effect of controlled-processing different semantic representations, the differences in the middle and late windows were confounded by emotion suppression. Therefore, our conclusions about the correlation between the regulatory effect of cognitive control and depression level and the role of rumination in mediating this correlation are limited to the early stage of emotion generation.

Extensive studies have revealed the relationships among cognitive factors, emotion regulation, and depressive symptoms [6]. We investigated the paths among these factors and found that the regulatory function of cognitive control influenced depression level via the mediation of rumination. This finding suggests that the decreased regulatory efficacy of cognitive control on emotion might contribute to the accumulation of negative mood, which facilitates rumination on mood and mood-congruent cognitions, thereby maintaining or intensifying depressive symptoms. Furthermore, it suggests that the impaired regulatory efficacy of cognitive control not only causes temporal elevated negative emotions but also promotes maladaptive emotion regulation in everyday life, which has long, sustained impact on mood and depressive symptoms.

## Conclusions

The present study demonstrated that the regulatory effect of controlled-processing different semantic representations on emotion varied with depression level. Furthermore, this study revealed that the regulatory efficacy of cognitive control influenced depression level via the mediation of rumination.

## Limitations

The present study has several limitations. First, most of the participants in our study were female; thus, whether the impaired cognitive control effect on emotion-related electrocortical response can be generalized to male participants with depressive symptoms remains unknown. Further studies needed to answer this question. Second, all the participants in our study were university students. Whether the findings can be replicated in clinically depressed individuals requires further investigation. Third, the anxiety state of the participants was not measured. There is comorbidity between depression and anxiety, and cognitive control deficit has been found in depressed patients with comorbid anxiety [57]. Whether the impaired regulatory effect of cognitive control in the high depression group was associated with the participants' anxiety state is unknown.

## Supporting information

**S1 Appendix. IAPS images and associated words.**
(XLSX)

**S2 Appendix. LPP amplitudes evoked by pictures presented for the first time.**
(SAV)

**S3 Appendix. LPP amplitudes evoked by pictures presented for the second time.**
(SAV)

## Acknowledgments

We gratefully thank Rang Yan and Bo Wu for help with data acquisition.

## Author Contributions

**Conceptualization:** Shuzhen Gan.

**Data curation:** Shuzhen Gan, Xiangrong Shen.

**Formal analysis:** Shuzhen Gan, Shuang Chen.

**Funding acquisition:** Shuzhen Gan.

**Investigation:** Shuzhen Gan.

**Methodology:** Shuzhen Gan.

**Project administration:** Shuzhen Gan, Xiangrong Shen.

**Resources:** Shuzhen Gan.

**Software:** Shuzhen Gan.

**Supervision:** Xiangrong Shen.

**Validation:** Shuang Chen.

**Visualization:** Shuzhen Gan.

**Writing – original draft:** Shuzhen Gan, Xiangrong Shen.

**Writing – review & editing:** Shuzhen Gan, Shuang Chen, Xiangrong Shen.

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
