## [Decision Letter · Decision Letter 0]

29 Aug 2019

PONE-D-19-20025

Impaired regulatory efficacy of controlled-processing semantic representations on emotion contributes to depressive symptoms: Evidences from an ERP study

PLOS ONE

Dear Dr. Gan,

Thank you for submitting your manuscript to PLOS ONE. After careful consideration, we feel that it has merit but does not fully meet PLOS ONE’s publication criteria as it currently stands. Therefore, we invite you to submit a revised version of the manuscript that addresses the points raised during the review process.

We would appreciate receiving your revised manuscript by Oct 13 2019 11:59PM. To enhance the reproducibility of your results, we recommend that if applicable you deposit your laboratory protocols in protocols.io, where a protocol can be assigned its own identifier (DOI) such that it can be cited independently in the future. For instructions see: http://journals.plos.org/plosone/s/submission-guidelines#loc-laboratory-protocols

We look forward to receiving your revised manuscript.

Kind regards,

Kai Wang

Academic Editor

PLOS ONE

Journal Requirements:

2. We note you have included a table to which you do not refer in the text of your manuscript. Please ensure that you refer to Table 4 in your text; if accepted, production will need this reference to link the reader to the Table.

Reviewers' comments:

Reviewer's Responses to Questions

**Comments to the Author**

1. Is the manuscript technically sound, and do the data support the conclusions?

Reviewer #1: Yes

Reviewer #2: Partly

2. Has the statistical analysis been performed appropriately and rigorously? 

Reviewer #1: Yes

Reviewer #2: Yes

3. Have the authors made all data underlying the findings in their manuscript fully available?

Reviewer #1: Yes

Reviewer #2: Yes

4. Is the manuscript presented in an intelligible fashion and written in standard English?

Reviewer #1: Yes

Reviewer #2: No

5. Review Comments to the Author

Reviewer #1: The topic of this study has some new ideas, but there are still some points that need to be clarified and modified.

1. The title of the paper will cause ambiguity, from which the research variables and topics cannot be clearly understood. The match between the topic and the key words is not high.

2. In participants section, ”Participants were paid 20 yuan after completing the SDS questionnaire and 80 yuan after the ERP experiment”. But there's nothing about the RRS for testing. When to test? How about payment?

3. Why not to use Emotion Image Library in Chinese Version?

Reviewer #2: The current study examines emotion regulation effect in participants with low and high depression level. The authors analyze behavioral and ERP data. They concluded that the underlying mechanisms of the link between the function of cognitive control in emotion generation and depressive symptoms, and indicated the pathway from the regulatory efficacy of cognitive control to depression via rumination.

The research question is interesting and has the potential to increase our understanding of emotion regulation in depression.

However, I have major concerns and I think that the data is not adequately analysed and interpreted and I cannot recommend publishing the paper in its current form.

I will outline my main concerns and comments below:

1. The title used ‘depressive symptom’. However, the participants in the current study were ones with low and high depressive level. Maybe the ‘depressive sate’ was proper.

2. The theoretical basis of this paper is based on patients with depression. However, ordinary college students with high SDS scores selected in this experiment can only be regarded as trait population, so whether the theoretical basis is universal in the study of trait population. Relevant literature is not listed for illustration.

3. The data analysis section, the authors use the regression methods to remove Vertical ocular artifacts. In fact, the ICA method was more reliable than regression procedure for artifacts removing. Why the authors don’t use the ICA method?

4. Previous studies have confirmed that theta-band oscillations are critical for cognitive control. I suggest that authors doing time-frequency analysis to explore the TF information in the data.

5. In the results section, what’s the mean of ‘total’ in the table 4. It was not correct to calculate the regression between all participants and LPP amplitudes, as all participants were not continuous variable.

6. The figure was not marked clearly. What’s the figure in three rows represent in Fig4 and Fig5? The statistical value should be written in Fig6.

7. In Table3, the early LPP amplitude in the high depression group and the low depression group showed significant differences under the conditions of cp-u and NEU. However, in the middle and late windows, the differences in the high depression group under the conditions of cp-u and NEU gradually disappeared, and the differences in the low depression group varied, which was not explained in the discussion.

8. The written English needs considerable improvement.

6. PLOS authors have the option to publish the peer review history of their article (what does this mean?). If published, this will include your full peer review and any attached files.

Reviewer #1: No

Reviewer #2: No

---

## [Author Response · Author response to Decision Letter 0]

27 Oct 2019

Dear editor and reviewers:

We are truly grateful to your comments and thoughtful suggestions on our manuscript (PONE-D-19-20025). Based on the comments we received, careful modifications have been made to the manuscript. Changes to our manuscript were all highlighted in the version with track-changes. Point-by-point responses to your comments are listed in the attached file "Response to reviewers". We thank you for the time and efforts on our paper. We look forward to hearing from you. 

Wish you all the best!

Sincerely yours,

Shuzhen Gan

---

## [Editor Report · Decision Letter 1]

1 Nov 2019

The emotion regulation effect of cognitive control is related to depressive state through the mediation of rumination: An ERP study

PONE-D-19-20025R1

Dear Dr. Gan,

We are pleased to inform you that your manuscript has been judged scientifically suitable for publication and will be formally accepted for publication once it complies with all outstanding technical requirements.

With kind regards,

Kai Wang

Academic Editor

PLOS ONE
---

## [Editor Report · Acceptance letter]

7 Nov 2019

PONE-D-19-20025R1 

The emotion regulation effect of cognitive control is related to depressive state through the mediation of rumination: An ERP study 

Dear Dr. Gan:

I am pleased to inform you that your manuscript has been deemed suitable for publication in PLOS ONE. Congratulations! Your manuscript is now with our production department. 

With kind regards,

on behalf of

Prof. Kai Wang 

Academic Editor

PLOS ONE